# COMBINING STRUCTURE AND TEXT: LEARNING REPRESENTATIONS FOR REASONING ON GRAPHS

## ABSTRACT

Effective reasoning on real-world graphs necessitates a thorough understanding and optimal utilization of structural information from graph structure and textual information corresponding to nodes and edges. Recent research has primarily focused on two paradigms: employing graph neural networks to capture structural features and utilizing language models to process textual information, respectively. While these approaches have shown impressive performance, integrating structural and textual information presents significant challenges. To be more specific, concurrently training graph neural networks and language models is particularly challenging, primarily due to the scale of real-world graphs. Additionally, the dynamic set of answer nodes poses a difficulty to the design of joint optimization objectives. This paper introduces a novel framework, named `CoST`, tailored for graph reasoning tasks. The proposed optimization objective enables alternating training of the GNN and PLM, leading to the generation of effective text representations by the PLM model, thereby enhancing the reasoning capabilities of the GNN model. Empirical results demonstrate that `CoST` achieves state-of-the-art performance across representative benchmark datasets.

## 1 INTRODUCTION

Significant progress has been achieved through the utilization of deep learning methods to enhance reasoning tasks in recent years. The realm of reasoning encompasses a diverse array of tasks, including natural language reasoning (Clark et al., 2020; Talmor et al., 2020; Wei et al., 2022; Suzgun et al., 2023), visual reasoning (Battaglia et al., 2016; Weston et al., 2016; Hu et al., 2017), multi-modal reasoning (Johnson et al., 2017; Nam et al., 2017; Cadène et al., 2019; Lu et al., 2022; Yang et al., 2023b), mathematical reasoning (Saxton et al., 2019; Zhang et al., 2019a; Lu et al., 2023; Yang et al., 2023a; Gou et al., 2024), and sophisticated hard science reasoning (Fragkiadaki et al., 2016; Bongini et al., 2021; Lu et al., 2022).

Graph-structured data plays a pivotal role in modeling complex and dynamic systems and the interconnections among system components (Zartman & Gordon, 1981; Pan et al., 2021; Das & Soylu, 2023), making it a crucial data type in reasoning tasks (Chen et al., 2019; Zhang et al., 2019b; 2021). Reasoning on graphs typically involves a query and aims to predict a set of nodes as answers. In recent years, a considerable direction of research has delved into enhancing reasoning capabilities on graphs. Traditional methods (Katz, 1953; Page, 1999; Borgatti & Everett, 2006) often rely on statistical data and heuristic metrics to extract graph data features for reasoning purposes. More recently, graph neural networks (GNNs) have demonstrated remarkable performance across various graph-related tasks, including graph reasoning. GNNs (Kipf & Welling, 2017; Velickovic et al., 2018; Xu et al., 2019) employ a message-passing mechanism that enables them to capture transferable invariances of structural nature among graphs, enhancing their ability to conduct reasoning tasks effectively. However, these studies still encounter the issue: of solely focusing on graph structure while overlooking non-structural information associated with nodes and edges in graphs. In this paper, we focus on the textual information embedded in the text descriptions corresponding to nodes and edges.

Consider Figure 1 as an illustration: given a heterogeneous graph representing persons and their associated teams, suppose the goal is to answer queries such as (Kylian Mbappé, belong, ?). This graph comprises nodes characterized as either *person* or *team*, connected by rela-

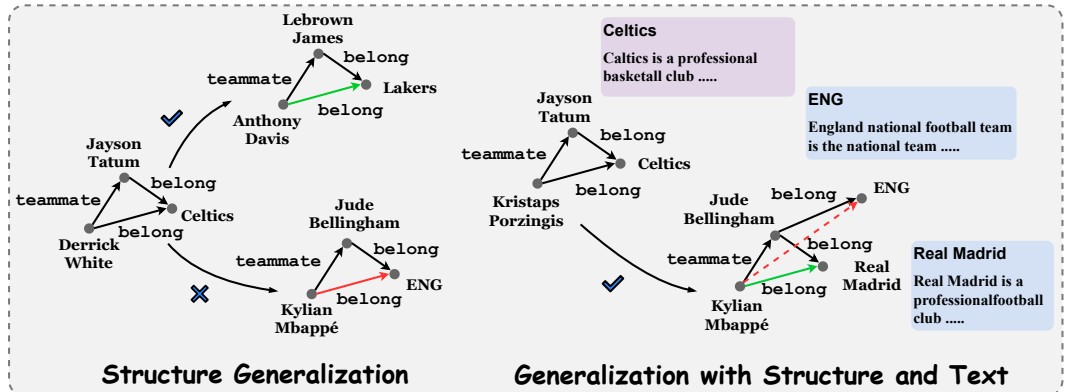

Figure 1: Illustration of reasoning on graphs using different information. ■ and ■ indicate the textual description corresponding to nodes.

tions, including *teammate* and *belong*. GNNs can detect patterns (Qiu et al., 2024), like *the association between two persons as teammates implying they belong to the same team.* While leveraging this learned pattern, GNNs can correctly respond to queries like (`Anthony Davis,belong,Lakers`). However, these methods may erroneously predict answers, as in the case of (`Kylian Mbappé,belong,ENG`). By incorporating textual data, models gain the ability to differentiate between various *team* nodes based on their descriptions. As shown in Figure 1, models can accurately predict (`Kylian Mbappé,belong,Real Madrid`) with textual information.

Despite the promising application, discrete text information presents a challenge for neural networks relying on stochastic gradient descent. An intuitive solution is to utilize a pre-trained language model (Devlin et al., 2019; Liu et al., 2019; Yang et al., 2019; Dong et al., 2019; Bao et al., 2020; He et al., 2021) (PLM) to convert textual data into continuous embeddings for reasoning or integration into graph reasoning models. Static text representations fall short in capturing contextual nuances (Peters et al., 2018), highlighting the need for fine-tuning the pre-trained language model for effective reasoning. However, the scale of graph data presents substantial challenges when attempting to fine-tune pre-trained language models in conjunction with reasoning GNN models (Zhao et al., 2023). In addition, for graph reasoning tasks, the probability of all candidate nodes on the graph needs to be modeled using pairwise representations under different query contexts. This requirement drives GNN-based models and PLM-based models to adopt different reasoning pipelines, thereby posing significant challenges to the interaction and co-optimization of these models.

To overcome the aforementioned challenges, we propose an assembled graph reasoning framework that effectively **Co**mbines **S**tructural and **T**extual information, named `CoST`. Specifically, we first introduce an objective to integrate both structural and textual elements into a GNN-based reasoning model coupled with a PLM model, utilized for producing text embeddings. Additionally, we formulate an alternative objective function enabling us to optimize the GNN and PLM models in a cyclical manner. Each iteration involves training both models using observed data from the training graph and pseudo target generated by a separate fixed model. We conduct extensive experiments on several benchmark datasets to demonstrate the superior performance of `CoST`. Significantly, our framework enhances the performance of the GNN model by offering effective text representations from the PLM model, leading to `CoST` achieving state-of-the-art performance on benchmarks.

## 2 METHODOLOGY

### 2.1 TASK FORMULATION

Given a graph $\mathcal{G} = \{\mathcal{V}, \mathcal{R}, \mathcal{E}\}$[1], the query $(h, r, ?)$ is linked to an answer set $\mathcal{A}_{(h,r)} \subseteq \mathcal{V}$, where $\mathcal{V}$ represents the node set, $\mathcal{R}$ denotes the relation set, and $\mathcal{E} \subseteq \mathcal{V} \times \mathcal{R} \times \mathcal{V}$ characterizes the set of

---

[1]This definition typically characterizes a heterogeneous graph (Sun et al., 2011; 2013); for homogeneous graphs, it can be viewed as a specific case where $|\mathcal{R}| = 1$.

edges. Here, $h \in \mathcal{V}$ and $r \in \mathcal{R}$ correspond to the query node and query relation, respectively. Generally, the sets $\mathcal{V}$ and $\mathcal{R}$ are associated with sequential text features such as sentences or paragraphs, allowing the graph to be defined as $\mathcal{G} = \{\mathcal{V}, \mathcal{R}, \mathcal{E}, \mathcal{T}_\mathcal{V}, \mathcal{T}_\mathcal{R}\}$. In this paper, we focus on the task of answering the query $(h, r, ?)$ by utilizing the structural information $\mathcal{E}$ and the textual information $\mathcal{T} = \{\mathcal{T}_\mathcal{V}, \mathcal{T}_\mathcal{R}\}$. Formally, we aim to model the distribution $p(t|h, r, \mathcal{E}, \mathcal{T})$ for each $t \in \mathcal{V}$, assessing the plausibility of $t \in \mathcal{A}_{(h,r)}$.

## 2.2 CoST: Model Design and Optimization Objective

Intuitively, the aforementioned problem can be effectively tackled by using graph neural networks (GNNs) in a message-passing manner (Schlichtkrull et al., 2018; Zhang et al., 2019c; Hu et al., 2020b; Vashishth et al., 2020; Zhu et al., 2021b; 2023). To be specific, the message-passing mechanism for a query $(h, r_q, ?)$ can be outlined as follows:

$$\boldsymbol{h}_u^{(0)} \leftarrow \text{INIT}(u|h, r_q),$$

$$\boldsymbol{h}_u^{(l+1)} \leftarrow \text{UPD}\left(\boldsymbol{h}_u^{(l)}, \text{AGG}\left(\left\{\left\{\text{MSG}\left(\boldsymbol{h}_v^{(l)}, \boldsymbol{e}_{r_q}\right) \mid v \in \mathcal{N}_r(u), r \in \mathcal{R}\right\}\right\}\right)\right), \quad (1)$$

$$p\left(t|h, r, \mathcal{E}_{/\{(h,r_q,t)\}}\right) \leftarrow \text{DEC}\left(\boldsymbol{h}_t^{(L)}\right),$$

where $\boldsymbol{h}_u$ and $\boldsymbol{e}_r$ indicates the representation for nodes and relations, respectively, INIT, MSG, AGG, UPD, and DEC represents the *initialization, message, aggregation, update*, and *decoder* functions, respectively and $\mathcal{N}_r(u)$ dedicates the neighbor nodes of $u$ corresponds to relation $r$. $\phi = \{\text{INIT}, \text{MSG}, \text{AGG}, \text{UPD}, \text{DEC}\}$ denotes the parameters of GNN models. Noticeably, most previous methods have typically utilized random initialization or fixed embeddings for *initialization* function Init, neglecting the textual information. As mentioned before, the discrete nature of textual information poses a challenge as it cannot be directly integrated into the GNN model. A straightforward approach is to implement INIT by leveraging a pre-trained language model (PLM) based *encoder* function (e.g., BERT (Devlin et al., 2019) and RoBERTa (Liu et al., 2019)) to encode textual information into continuous embeddings:

$$\boldsymbol{h}_u^{(0)} \leftarrow \text{INIT}(u|h, r_q, \text{ENC}(\mathcal{T}_u)), \quad \boldsymbol{e}_r \leftarrow \text{ENC}(\mathcal{T}_r),$$

$$p\left(t|h, r, \mathcal{E}_{/\{(h,r_q,t)\}}, \mathcal{T}\right) \leftarrow \text{DEC}\left(\boldsymbol{h}_t^{(L)}\right). \quad (2)$$

Subsequently, $\phi = \{\text{INIT}, \text{MSG}, \text{AGG}, \text{UPD}, \text{ENC}, \text{DEC}\}$ can be updated by maximizing the following objective:

$$\mathcal{O}(\phi) = \sum_{\mathcal{S}} \log p_\phi\left(O_{(h,r)}|h, r, \mathcal{E}_{/\{(h,r,\hat{t})\}_{\hat{t} \in O_{(h,r)}}}, \mathcal{T}\right), \quad (3)$$

where $\mathcal{S} = \{(h, r)|(h, r, t) \in \mathcal{E}, t \in \mathcal{V}\}$ and $O_{(h,r)}$ denotes the observed answer node set corresponding to the query $(h, r, ?)$ in the training data.

However, optimizing the pre-trained language model based encoder ENC and the GNN model jointly presents challenges due to the immense scale of real-world graphs, characterized by a large number of nodes, relations, and edges. To mitigate this challenge, we introduce a variational distribution $q_\theta\left(H_{(h,r)}|h, r, \mathcal{T}\right)$, where $\theta$ representing the associated parameters. In this context, $H_{(h,r)} = \mathcal{V}_{/O_{(h,r)}}$ indicates the candidate target node set for the query $(h, r, ?)$ that includes the hidden answer nodes. Notice that $q$ is solely conditioned on the textual information $\mathcal{T}$, enabling the parameterization of $q_\theta\left(H_{(h,r)}|h, r, \mathcal{T}\right)$ using a PLM that shares parameters with ENC.

**Parameterization of $q_\theta$.** In parameterizing the distribution $q$, we opt to utilize the same PLM employed for generating embeddings for INIT function of the GNN model. Specifically, we encode the textual descriptions of $h$, $r$, and $\tilde{t} \in H_{(h,r)}$ using the PLM and subsequently compute the distribution $q_\theta\left(\tilde{t}|h, r, \mathcal{T}\right)$ by an decoder network $g$:

$$q_\theta\left(\tilde{t}|h, r, \mathcal{T}\right) = g\left(\text{PLM}(\mathcal{T}_h), \text{PLM}(\mathcal{T}_r), \text{PLM}(\mathcal{T}_{\tilde{t}})\right). \quad (4)$$

Also, we hypothesize $q_\theta\left(\tilde{t}|h, r, \mathcal{T}\right)$ is only relates to $\mathcal{T}_h$, $\mathcal{T}_r$, and $\mathcal{T}_{\tilde{t}}$, leading to the formulation (Kadanoff, 2009; Kingma & Welling, 2014):

$$q_\theta(H_{(h,r)}|h, r, \mathcal{T}) = \prod_{\tilde{t} \in H_{(h,r)}} q_\theta(\tilde{t}|h, r, \mathcal{T}). \quad (5)$$

---

**Algorithm 1:** COST Framework

---

**Input:** PLM model $\theta$, GNN model $\phi$, Graph $\mathcal{G} = \{\mathcal{V}, \mathcal{R}, \mathcal{E}, \mathcal{T}_\mathcal{V}, \mathcal{T}_\mathcal{R}\}$, Update steps $L$
**Output:** Trained PLM model $\overline{\theta}$, Trained GNN model $\overline{\phi}$
/* Pre-training PLM model                                                    */
**while** *Not convergence or reach the max training epochs* **do**
  $\quad$ Obtain $\widehat{\theta}$ by optimizing $\sum_\mathcal{S} q_\theta \left( O_{(h,r)} | h, r, \mathcal{T} \right)$;
**end**
/* Pre-training GNN model                                                    */
**while** *Not convergence or reach the max training epochs* **do**
  $\quad$ Obtain $\widehat{\phi}$ by optimizing $\sum_\mathcal{S} p_\phi \left( O_{(h,r)} | h, r, \mathcal{E}_{/\{(h,r,\hat{t})\}_{\hat{t} \in O_{(h,r)}}}, \mathcal{T} \right)$;
**end**
Assign $(\widehat{\theta}, \widehat{\phi})$ to $(\overline{\theta}_0, \overline{\phi}_0)$ ;
**for** $i \leftarrow 1$ **to** $L$ **do**
  $\quad$ Fix GNN model $\overline{\phi}_{i-1}$ and update PLM model $\overline{\theta}_{i-1}$ to obtain $\overline{\theta}_i$ by optimizing $\mathcal{O}_{\text{PLM}}$ in
  $\quad$ Equation (9) ;
  $\quad$ Fix PLM model $\overline{\theta}_{i-1}$ and update GNN model $\overline{\phi}_{i-1}$ to obtain $\overline{\phi}_i$ by optimizing $\mathcal{O}_{\text{GNN}}$ in
  $\quad$ Equation (12) ;
**end**

---

The introduction of $q$ leads us to the following theorem:

**Theorem 2.1.** *Optimizing the objective function* $\mathcal{O}(\phi)$ *in Equation* (3) *is equivalent to optimizing the following objective:*

$$
\mathcal{O}(\phi, \theta) = \sum_\mathcal{S} \mathbb{E}_{q_\theta\left(H_{(h,r)} | h, r, \mathcal{T}\right)} \underbrace{\left[ \log \frac{p_\phi \left( O_{(h,r)}, H_{(h,r)} | h, r, \mathcal{E}_{/\{(h,r,\hat{t})\}_{\hat{t} \in O_{(h,r)}}}, \mathcal{T} \right)}{q_\theta \left( H_{(h,r)} | h, r, \mathcal{T} \right)} \right]}_{\text{Evidence Lower Bound}} +
$$
$$
\underbrace{D_{\text{KL}} \left( q_\theta \left( H_{(h,r)} | h, r, \mathcal{T} \right) \| p_\phi \left( H_{(h,r)} | h, r, \mathcal{E}, \mathcal{T} \right) \right)}_{\text{KL divergence}},
$$
(6)

*where* $p_\phi \left( O_{(h,r)}, H_{(h,r)} | h, r, \mathcal{E}_{/\{(h,r,\hat{t})\}_{\hat{t} \in O_{(h,r)}}}, \mathcal{T} \right)$ *represents the joint distribution,* $p_\phi \left( H_{(h,r)} | h, r, \mathcal{E}, \mathcal{T} \right)$ *denotes the posterior distribution,* $q_\theta \left( H_{(h,r)} | h, r, \mathcal{T} \right)$ *illustrates the variational distribution.*

The proof is provided in Appendix B. Based on Theorem 2.1, the optimization of the objective in Equation (6) can be implemented as an alternating process between optimizing the distributions $p$ and $q$ (Neal & Hinton, 1998; McLachlan & Krishnan, 2007; Qu et al., 2019; Zhao et al., 2023). The optimization of $q$ focuses on reducing the KL divergence between *variational distribution* and *posterior distribution* to enhance the tightness of the lower bound while optimizing $p$ aims to maximize the evidence lower bound (ELBO) to optimize the overall objective.

The upcoming sections will elaborate on the optimization strategies employed to facilitate collaboration between the PLM and GNN models.

### 2.3 COST: OPTIMIZATION FRAMEWORK

#### 2.3.1 PLM OPTIMIZATION

During the optimization of the PLM, the GNN model remains fixed while the PLM model is updated by minimizing the KL divergence between the *posterior distribution* and *variational distribution*. To be specific, we aim to minimize $D_{\text{KL}} \left( q_\theta \left( H_{(h,r)} | h, r, \mathcal{T} \right) \| p_\phi \left( H_{(h,r)} | h, r, \mathcal{E}, \mathcal{T} \right) \right)$, which is equiv-

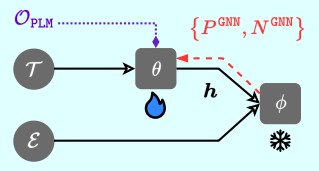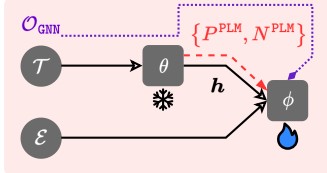

Figure 2: Illustration of ⬛ for COST framework, ⬛ optimization of PLM model, and ⬛ for optimization of GNN model, where 💧 indicates the trainable model and ❄ indicates the fixed model.

alent to minimizing:

$$q_\theta \left( H_{(h,r)}|h,r,\mathcal{T} \right) \log q_\theta \left( H_{(h,r)}|h,r,\mathcal{T} \right) - q_\theta \left( H_{(h,r)}|h,r,\mathcal{T} \right) \log p_\phi \left( H_{(h,r)}|h,r,\mathcal{E},\mathcal{T} \right). \quad (7)$$

However, optimizing the mentioned objective proves challenging because of the intricateness and instability of optimizing the entropy of $q_\theta$, which is shown in the first term.

Note that $p_\phi \left( H_{(h,r)}|h,r,\mathcal{E},\mathcal{T} \right)$ represents the prediction of the GNN model to the candidate target node set and remains unchanged due to the fixed GNN model. We can alternatively optimize the PLM model with the utilization of *hard pseudo target* produced by the GNN model. Formally, we obtain pseudo target $\left\{ P_{(h,r)}^{\text{GNN}}, N_{(h,r)}^{\text{GNN}} \right\}$ from $p_\phi \left( H_{(h,r)}|h,r,\mathcal{E},\mathcal{T} \right)$ by:

$$P_{(h,r)}^{\text{GNN}} \sim \mathcal{M} \left( \left\{ p_\phi \left( \tilde{t}|h,r,\mathcal{E},\mathcal{T} \right) \right\}_{\tilde{t} \in H_{(h,r)}}, \left| H_{(h,r)} \right| \right),$$

$$N_{(h,r)}^{\text{GNN}} \sim \mathcal{M} \left( \left\{ \overline{p}_\phi \left( \tilde{t}|h,r,\mathcal{E},\mathcal{T} \right) \right\}_{\tilde{t} \in H_{(h,r)}}, \left| H_{(h,r)} \right| \right), \quad (8)$$

where $\mathcal{M}$ means the multinomial distribution. Intuitively, $N_{(h,r)}^{\text{GNN}} \in H_{(h,r)}$ indicates the negative target and $P_{(h,r)}^{\text{GNN}} \in H_{(h,r)}$ indicates the positive target for query $(h,r,?)$ predicted by the fixed GNN model, respectively. We combine $P_{(h,r)}^{\text{GNN}}$ with $O_{(h,r)}$ to obtain $\widehat{P}_{(h,r)}^{\text{GNN}}$, and the objective for the PLM model can be described as:

$$\mathcal{O}_{\text{PLM}} = \sum_{\mathcal{S}} \left( \sum_{\hat{t} \in \widehat{P}_{(h,r)}^{\text{GNN}}} \frac{\exp \left( q_\theta \left( \hat{t}|h,r,\mathcal{T} \right) \right)}{\exp \left( q_\theta \left( \hat{t}|h,r,\mathcal{T} \right) \right) + \sum_{\tilde{t} \in N_{(h,r)}^{\text{GNN}}} \exp \left( q_\theta \left( \tilde{t}|h,r,\mathcal{T} \right) \right)} \right). \quad (9)$$

Intuitively, we utilize both the observed answer node set and the predicted pseudo target node set to optimize the PLM model by a contrastive loss.

### 2.3.2 GNN OPTIMIZATION

While optimizing the GNN model, the parameters of the PLM model remain constant, with the GNN model being updated through the maximization of the evidence lower bound. In particular, we leverage the fixed PLM to generate the continuous embeddings for textual information $\{\mathcal{T}_\mathcal{V}, \mathcal{T}_\mathcal{R}\}$ by $h_\mathcal{V} \leftarrow \text{PLM}(\mathcal{T}_\mathcal{V})$ and $e_\mathcal{R} \leftarrow \text{PLM}(\mathcal{T}_\mathcal{R})$, which are then feed into the GNN model. Then, we aim to optimize:

$$\mathbb{E}_{q_\theta \left( H_{(h,r)}|h,r,\mathcal{T} \right)} \left[ \log p_\phi \left( O_{(h,r)}, H_{(h,r)}|h,r,\mathcal{E}_{/\{(h,r,\hat{t})\}_{\hat{t} \in O_{(h,r)}}}, \mathcal{T} \right) \right]$$

$$= \mathbb{E}_{q_\theta \left( H_{(h,r)}|h,r,\mathcal{T} \right)} \left[ \log \sum_{t \in \mathcal{V}} p_\phi \left( t|h,r,\mathcal{E}_{/\{(h,r,\hat{t})\}_{\hat{t} \in O_{(h,r)}}}, \mathcal{T} \right) \right]. \quad (10)$$

Inspired by previous studies (Mikolov et al., 2013; Tang et al., 2015; Qu et al., 2019; Zhao et al., 2023), we approximate the optimization process by sampling $\left\{ P_{(h,r)}^{\text{PLM}}, N_{(h,r)}^{\text{PLM}} \right\}$ from $q_\theta$ and leveraging them to update the GNN model The acquisition of $\left\{ P_{(h,r)}^{\text{PLM}}, N_{(h,r)}^{\text{PLM}} \right\}$ mirrors the manner as shown in Equation (8):

$$P_{(h,r)}^{\text{PLM}} \sim \mathcal{M} \left( \left\{ q_\theta \left( \tilde{t}|h,r,\mathcal{T} \right) \right\}_{\tilde{t} \in H_{(h,r)}}, \left| H_{(h,r)} \right| \right),$$

$$N_{(h,r)}^{\text{PLM}} \sim \mathcal{M} \left( \left\{ \overline{q}_\theta \left( \tilde{t}|h,r,\mathcal{T} \right) \right\}_{\tilde{t} \in H_{(h,r)}}, \left| H_{(h,r)} \right| \right). \quad (11)$$

By combining $\left\{ P_{(h,r)}^{\text{PLM}}, N_{(h,r)}^{\text{PLM}} \right\}$ with the observed answer node set $O_{(h,r)}$, we establish the objective for updating the GNN model:

$$\mathcal{O}_{\text{GNN}} = \sum_{\mathcal{S}} \left( \gamma \frac{1}{\left| N_{(h,r)}^{\text{PLM}} \right|} \sum_{\tilde{t} \in N_{(h,r)}^{\text{PLM}}} \log \left( 1 - p_\phi \left( \tilde{t}|h, r, \mathcal{E}_{/\{(h,r,\hat{t})\}_{\hat{t} \in O_{(h,r)}}}, \mathcal{T} \right) \right) \right) +$$

$$\sum_{\mathcal{S}} \left( \tau \frac{1}{\left| \widehat{P}_{(h,r)}^{\text{PLM}} \right|} \sum_{\hat{t} \in \widehat{P}_{(h,r)}^{\text{PLM}}} \log p_\phi \left( \hat{t}|h, r, \mathcal{E}_{/\{(h,r,\hat{t})\}_{\hat{t} \in O_{(h,r)}}}, \mathcal{T} \right) \right), \quad (12)$$

where $\widehat{P}_{(h,r)}^{\text{PLM}} = \{ P_{(h,r)}^{\text{PLM}}, O_{(h,r)} \}$ and $(\gamma, \tau)$ are hyperparameters controlling different term weights.

### 2.4 CoST: A High-level Summary

The high-level illustration of the CoST is delineated in Figure 2 and Algorithm 1. Initially, a PLM model and a GNN model are integrated to form the CoST framework. Then, to expedite convergence and enhance model performance, the GNN model and PLM model are pre-trained on a certain dataset. The subsequent training iterates between the GNN and PLM models, adhering to the objective functions specified in Equation (9) and Equation (12) to finalize the training of CoST.

## 3 Experiment

### 3.1 Experiment on Homogeneous Graphs

#### 3.1.1 Experimental Settings

**Datasets.** We utilize four datasets, namely AmazonSports (McAuley et al., 2015), AmazonClothing (McAuley et al., 2015), MAGGeology (Zhang et al., 2023b), and MAGMath (Zhang et al., 2023b) introduced by He et al. (2024b) to assess the performance of methods on the homogeneous graph reasoning task. Furthermore, the large-scale datasets CitationV8 and GoodReads, introduced by Yan et al. (2023), are employed as benchmarks for evaluating the effectiveness on the larger homogeneous graphs. Table 1 illustrates the dataset statistics and further details about datasets are available in Appendix C.1.

**Baselines.** For the AmazonSports, AmazonClothing, MAGGeology, and MAGMath datasets, we consider two categories of baselines: 1) GNN-based methods, including GCN (Kipf & Welling, 2017), GraphSAGE (Hamilton et al., 2017), and GATv2 (Brody et al., 2022); 2) LLM-based methods, such as GraphGPT (Tang et al., 2024), LLaGA (Chen et al., 2024) and LINKGPT (He et al., 2024b). For the GNN-based methods, we use the embeddings produced by BERT (Devlin et al., 2019) following original papers. Applying large language model methods to large-scale datasets

Table 1: Dataset statistics of homogeneous datasets and heterogeneous datasets.

| Dataset | #Node | #Relation | #Train | #Valid | #Test |
|---|---|---|---|---|---|
| **AmazonSports** | $20,417$ | – | $40,297$ | $2,238$ | $2,240$ |
| **AmazonClothing** | $20,180$ | – | $46,187$ | $2,565$ | $2,567$ |
| **MAGGeology** | $20,530$ | – | $46,386$ | $2,577$ | $2,577$ |
| **MAGMath** | $19,878$ | – | $31,208$ | $1,733$ | $1,735$ |
| **CitationV8** | $1,106,759$ | – | $6,059,687$ | $30,604$ | $30,604$ |
| **CoodReads** | $676,084$ | – | $7,724,075$ | $171,646$ | $686,585$ |
| **FB15k237** | $14,541$ | $237$ | $272,115$ | $17,535$ | $20,466$ |
| **WN18RR** | $40,943$ | $11$ | $86,835$ | $3,034$ | $3,134$ |
| **Wikidata5M$_{\text{trans}}$** | $4,594,485$ | $822$ | $20,614,279$ | $5,163$ | $5,163$ |

Table 2: Results of CoST and other baseline models on the AmazonSports, AmazonClothing, MAGGeology, and MAGMath datasets. Red indicates the best results, green indicates the second-best results, and blue indicates improvement comparing with the pretrained GNN model.

| Method | AmazonSports | | AmazonClothing | | MAGGeology | | MAGMath | |
|---|---|---|---|---|---|---|---|---|
| | MRR↑ | H@1↑ | MRR↑ | H@1↑ | MRR↑ | H@1↑ | MRR↑ | H@1↑ |
| *GNN-based Method* | | | | | | | | |
| GCN (Kipf & Welling, 2017) | 70.4 | 60.2 | 68.2 | 60.0 | 51.4 | 40.1 | 45.8 | 33.1 |
| GraphSAGE (Hamilton et al., 2017) | 77.6 | 68.4 | 81.2 | 71.6 | 51.0 | 36.8 | 44.2 | 28.4 |
| GATv2 (Brody et al., 2022) | 81.4 | 73.0 | 87.8 | 81.6 | 65.7 | 55.3 | 51.6 | 38.0 |
| *LLM-based Method* | | | | | | | | |
| LLaMA2 (Touvron et al., 2023) | 40.8 | 30.9 | 30.2 | 22.5 | 22.9 | 13.5 | 21.9 | 13.5 |
| GraphGPT (Tang et al., 2024) | 14.8 | 6.0 | 32.3 | 14.3 | 12.4 | 4.4 | 9.8 | 2.6 |
| LLaGA (Chen et al., 2024) | 83.4 | 75.4 | 84.5 | 77.5 | 74.3 | 63.3 | 62.2 | 49.8 |
| LINKGPT (He et al., 2024b) | 87.1 | 79.6 | 90.2 | 84.8 | 81.0 | 71.0 | 75.4 | 64.6 |
| CoST $_{Pretrained\ PLM}$ | 56.2 | 40.1 | 61.1 | 48.1 | 20.9 | 11.3 | 29.0 | 17.5 |
| CoST $_{Pretrained\ GNN}$ | 85.9 | 78.5 | 88.7 | 83.6 | 80.1 | 69.8 | 74.3 | 62.9 |
| CoST | 88.3$_{\uparrow2.4}$ | 80.6$_{\uparrow2.1}$ | 91.4$_{\uparrow2.7}$ | 85.2$_{\uparrow1.6}$ | 82.3$_{\uparrow2.2}$ | 72.4$_{\uparrow2.6}$ | 76.6$_{\uparrow2.3}$ | 65.9$_{\uparrow3.0}$ |

Table 3: Results of CoST and other baseline models on the CitationV8 and GoodReads datasets. Red indicates the best results, green indicates the second-best results, and blue indicates improvement concerning the pretrained GNN model within CoST.

| Method | CitationV8 | | | | GoodReads | | | |
|---|---|---|---|---|---|---|---|---|
| | MRR↑ | H@10↑ | H@50↑ | H@100↑ | MRR↑ | H@10↑ | H@50↑ | H@100↑ |
| *GNN-based Method* | | | | | | | | |
| GCN (Kipf & Welling, 2017) | 60.0 | 50.4 | 75.1 | 90.2 | 65.1 | 55.4 | 85.0 | 91.5 |
| SAGE (Hamilton et al., 2017) | 54.0 | 44.1 | 71.3 | 89.1 | 65.7 | 54.1 | 82.9 | 89.6 |
| *PLM-based Method* | | | | | | | | |
| BERT$_{tiny}$ (Jiao et al., 2020) | 41.2 | 33.6 | 48.2 | 66.6 | 42.2 | 36.9 | 52.5 | 76.2 |
| BERT$_{base}$ (Devlin et al., 2019) | 44.6 | 38.9 | 57.5 | 72.4 | 44.4 | 44.0 | 60.9 | 79.2 |
| *Topological Contrastive Learning based Method* | | | | | | | | |
| GCN (Kipf & Welling, 2017) | 70.2 | 68.3 | 84.6 | 93.7 | 85.1 | 73.9 | 92.8 | 95.9 |
| SAGE (Hamilton et al., 2017) | 60.2 | 62.4 | 80.6 | 92.6 | 82.2 | 75.2 | 90.7 | 94.0 |
| BERT$_{tiny}$ (Jiao et al., 2020) | 47.3 | 41.3 | 57.3 | 72.6 | 55.1 | 45.5 | 61.6 | 82.4 |
| BERT$_{base}$ (Devlin et al., 2019) | 52.8 | 46.6 | 65.8 | 72.5 | 61.2 | 52.6 | 66.0 | 85.6 |
| CoST $_{Pretrained\ PLM}$ | 42.7 | 35.9 | 54.3 | 69.9 | 43.8 | 42.5 | 68.1 | 78.8 |
| CoST $_{Pretrained\ GNN}$ | 71.1 | 68.1 | 85.9 | 94.2 | 84.7 | 74.7 | 92.3 | 94.6 |
| CoST | 72.7$_{\uparrow1.6}$ | 69.5$_{\uparrow1.4}$ | 87.4$_{\uparrow1.5}$ | 95.2$_{\uparrow1.0}$ | 86.4$_{\uparrow1.7}$ | 76.1$_{\uparrow1.4}$ | 93.9$_{\uparrow1.6}$ | 97.0$_{\uparrow2.4}$ |

like CitationV8 and GoodReads proves challenging. Therefore, in such cases, we adopt BERT (Devlin et al., 2019) as the text-based baseline. Additionally, we incorporate topological contrastive learning as presented by Yan et al. (2023) as a strong baseline method that leverages both structural and textual information.

**Metrics.** We employ standard evaluation metrics (Wang et al., 2017; Hu et al., 2020a), including mean reciprocal rank (MRR) and Hits@$n$ ($n = 1, 10, 50, 100$). MRR represents the average reciprocal rank of all answer entities, while Hits@$n$ quantifies the percentage of answer entities ranked within the top-k positions.

### 3.1.2 RESULTS AND ANALYSIS

Table 2 and Table 3 illustrates the experimental results of CoST and baselines on the homogeneous graph reasoning datasets. Upon analysis of the results, it is evident that CoST outperforms prominent text-based and structure-based baselines across all datasets. Particularly on the Amazon and MAG datasets, CoST showcases superior performance compared to the resource-intensive LLM-based strategy. In the context of the CitationV8 and GoodReads datasets, our method outperforms models employing topological contrastive learning techniques. As mentioned before, this method can be viewed as a strong baseline that integrates both textual and structural information. The obser-

Table 4: Results of CoST and other baseline models on the FB15k237 and WN18RR datasets. Red indicates the best results, green indicates the second-best results, and blue indicates improvement concerning the pretrained GNN model.

| Method | FB15k237 | | | | WN18RR | | | |
|---|---|---|---|---|---|---|---|---|
| | MRR↑ | H@1↑ | H@3↑ | H@10↑ | MRR↑ | H@1↑ | H@3↑ | H@10↑ |
| *Embedding-based Method* | | | | | | | | |
| TransE (Bordes et al., 2013) | 27.9 | 19.8 | 37.6 | 44.1 | 24.3 | 4.3 | 44.1 | 53.2 |
| DistMult (Yang et al., 2015) | 28.1 | 19.9 | 30.1 | 44.6 | 44.4 | 41.2 | 47.0 | 50.4 |
| RotatE (Sun et al., 2019) | 33.8 | 24.1 | 37.5 | 53.3 | 47.6 | 42.8 | 49.2 | 57.1 |
| TuckER (Balazevic et al., 2019) | 35.8 | 26.6 | 39.4 | 54.4 | 47.0 | 44.3 | 48.2 | 52.6 |
| HousE (Li et al., 2022) | 36.1 | 26.6 | 39.9 | 55.1 | 51.1 | 46.5 | 52.8 | 60.2 |
| *Textual Information based Method* | | | | | | | | |
| MTL (Kim et al., 2020) | 26.7 | 17.2 | 29.8 | 45.8 | 33.1 | 20.3 | 38.3 | 59.7 |
| StAR (Wang et al., 2021a) | 29.6 | 20.5 | 32.2 | 48.2 | 40.1 | 24.3 | 49.1 | 70.9 |
| HittER (Chen et al., 2021) | 37.3 | 27.9 | 40.9 | 55.8 | 50.3 | 46.2 | 51.6 | 58.4 |
| *Structural Information based Method* | | | | | | | | |
| CompGCN (Vashishth et al., 2020) | 35.5 | 26.4 | 39.0 | 53.5 | 47.9 | 44.3 | 49.4 | 54.6 |
| NBFNet (Zhu et al., 2021b) | 41.5 | 32.1 | 45.4 | 59.9 | 55.1 | 49.7 | 57.3 | 66.6 |
| RED-GNN (Zhang & Yao, 2022) | 37.4 | 28.3 | – | 55.8 | 53.3 | 48.5 | – | 62.4 |
| A*Net (Zhu et al., 2023) | 41.1 | 32.1 | 45.3 | 58.6 | 54.9 | 49.5 | 57.3 | 65.9 |
| AdaProp (Zhang et al., 2023a) | 41.7 | 33.1 | – | 58.5 | 56.2 | 49.9 | – | 67.1 |
| CoST Pretrained PLM | 28.7 | 23.3 | 34.7 | 49.3 | 51.1 | 44.5 | 50.8 | 57.6 |
| CoST Pretrained GNN | 40.8 | 32.1 | 44.3 | 58.8 | 54.9 | 48.7 | 57.2 | 66.0 |
| CoST | 42.7↑1.9 | 34.5↑2.4 | 46.7↑2.4 | 60.9↑1.6 | 56.8↑1.9 | 51.0↑2.3 | 58.2↑1.0 | 67.4↑1.4 |

vation strongly suggests that our method efficiently utilizes both textual and structural data for graph reasoning tasks. Furthermore, comparing CoST with the GNN model at various stages (pretraining and after co-training) underscores the effectiveness and necessity of retraining the pre-trained language model for enhanced text representation.

## 3.2 EXPERIMENT ON HETEROGENEOUS GRAPHS

### 3.2.1 EXPERIMENTAL SETTINGS

**Datasets.** For the heterogeneous graph task, we focus on the knowledge graph reasoning datasets, including FB15k-237 (Toutanova & Chen, 2015), WN18RR (Saxe et al., 2014), and Wikidata5M (Wang et al., 2021c). Among them, FB15k237 and WN18RR are widely used knowledge graph inference datasets, while Wikidata5M is a large-scale knowledge graph dataset containing 5M entities. Wikidata5M provides data sets with transductive and inductive settings. However, the dataset with inductive setting does not provide

Table 5: Results of CoST and other baseline models on the Wikidata5M$_{trans}$ datasets.

| Method | Wikidata5M$_{trans}$ | | | |
|---|---|---|---|---|
| | MRR↑ | H@1↑ | H@3↑ | H@10↑ |
| *Embedding-based Method* | | | | |
| TransE (Bordes et al., 2013) | 25.3 | 17.0 | 31.1 | 39.2 |
| RotatE (Sun et al., 2019) | 29.0 | 23.4 | 32.2 | 39.0 |
| *Textual Information based Method* | | | | |
| SimKGC (Wang et al., 2022) | 35.8 | 31.3 | 37.6 | 44.1 |
| KGT5-context (Kochsiek et al., 2023) | 42.6 | 40.6 | 44.0 | 46.0 |
| *Structural Information based Method* | | | | |
| CompGCN (Vashishth et al., 2020) | 35.5 | 26.4 | 39.0 | 53.5 |
| CoST Pretrained PLM | 32.4 | 29.8 | 35.6 | 41.3 |
| CoST Pretrained GNN | 41.9 | 39.4 | 45.4 | 55.8 |
| CoST | 45.0↑3.1 | 43.7↑4.3 | 46.3↑0.9 | 57.4↑1.6 |

graph structural information, so the experiment is only carried out on Wikidata5M$_{trans}$. To further access the inductive performance, we involve fully inductive datasets proposed in Lee et al. (2023). More details about datasets can be found in Table 1 and Appendix C.1.

**Metrics.** Similar to the evaluation on the homogeneous datasets, we leverage MRR and Hits@$n$ ($n = 1, 3, 10$) as the evaluation metrics.

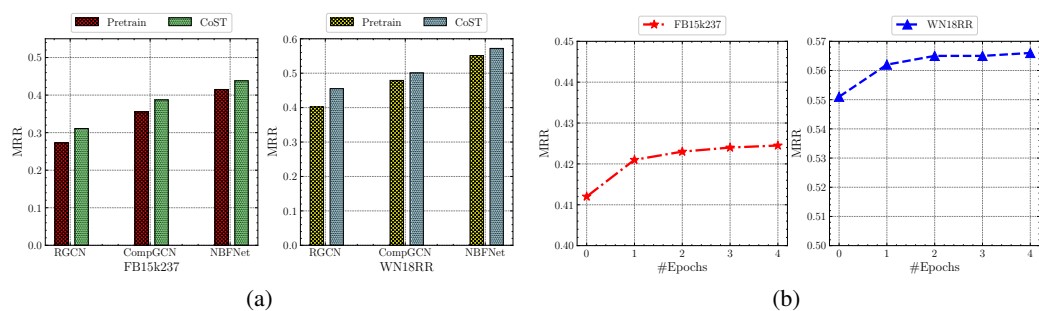

Figure 3: Illustration of ablation studies. (a) Ablation study of various GNN architectures - the left diagram displays the results for FB15k237, while the right diagram depicts the results for WN18RR. (b) Ablation study focusing on the convergence of `CoST` training, conducted on FB15k237 and WN18RR datasets.

**Baselines.** We employ three categories of baselines: 1) embedding-based methods, including TransE (Bordes et al., 2013), DistMult (Yang et al., 2015), RotatE (Sun et al., 2019), TuckER (Balazevic et al., 2019), and HousE (Li et al., 2022); 2) text-based methods, such as MTL (Kim et al., 2020), StAR (Wang et al., 2021a), SimKGC (Wang et al., 2022), HittER (Chen et al., 2021), and KGT5-context (Kochsiek et al., 2023); 3) strcture-based methods, which includes CompGCN (Vashishth et al., 2020), NBFNet (Zhu et al., 2021b), RED-GNN (Zhang & Yao, 2022), A*Net (Zhu et al., 2023), and AdaProp (Zhang et al., 2023a).

### 3.2.2 RESULTS AND ANALYSIS

The experimental results on the FB15k237 and WN18RR datasets are presented in Table 4. It is noteworthy that baseline models relying on structural information outperform those based on textual information, emphasizing the significance of structural data in knowledge graph reasoning tasks. Although the pretrained GNN model in `CoST` falls slightly behind key baselines, this discrepancy may be attributed to the difficulty of unaltered language models in providing effective text representations relevant to the task. However, following the alternating training of the GNN and PLM models, `CoST` surpasses all baselines on both datasets, underscoring the effectiveness of our approach. Furthermore, Table 5 demonstrates the results on the larger Wikidata5M dataset, where `CoST` also outperforms prominent baseline models.

### 3.3 ABLATION STUDIES

**Architectures.** In order to assess the generalizability of the proposed method, we ablation experiments involving different graph neural network architectures on the FB15k237 and WN18RR datasets. We specifically utilize RGCN (Schlichtkrull et al., 2018), CompGCN (Davidson et al., 2018), and NBFNet (Zhu et al., 2021b) as the backbone graph neural network architectures. The results depicted in Figure 3a demonstrate a consistent enhancement achieved by our approach across diverse graph neural network architectures.

**Training Paradigm.** To access the efficacy of the alternate training paradigm in `CoST`, we conducted experiments comparing different training methods. We specifically employed two paradigms: 1) *static*, generating text representations using a fixed pre-trained language model; and 2) *two-stage*, involving fine-tuning the pre-trained language model before text representation. The results shown in Table 6 demonstrate that `CoST` surpasses the baselines, confirming its effectiveness.

**Convergence.** Ensuring convergence when training with alternating paradigms is essential. To confirm the convergence of `CoST`, we conducted experiments on the FB15k237 and WN18RR datasets. The results depicted in Figure 3b showcase the model's rapid convergence, even during short training periods.

Table 6: Results of ablation study w.r.t. different training paradigm. Red indicates the best results and green indicates the second-best results.

| Dataset | Static | | | | Two-Stage | | | | CoST | | | |
|---|---|---|---|---|---|---|---|---|---|---|---|---|
| | MRR↑ | H@1↑ | H@3↑ | H@10↑ | MRR↑ | H@1↑ | H@3↑ | H@10↑ | MRR↑ | H@1↑ | H@3↑ | H@10↑ |
| AmazonSports | 85.8 | 76.4 | – | – | 86.9 | 78.8 | – | – | 88.3 | 80.6 | – | – |
| FB15k237 | 41.0 | 32.5 | 44.6 | 58.2 | 41.7 | 32.3 | 44.8 | 59.0 | 42.7 | 34.5 | 46.7 | 60.9 |

## 4 RELATED WORK

**Structure-based Methods.** Traditional methods for graph reasoning (Katz, 1953; Page, 1999; Borgatti & Everett, 2006) historically relied on statistical data and heuristic metrics to extract features from graph data for reasoning purposes. Recently, the widespread adoption of graph neural networks (GNNs) has enabled the learning of structural representations by encoding graph topologies. Prominent frameworks (Kipf & Welling, 2016; Schlichtkrull et al., 2018; Davidson et al., 2018; Vashishth et al., 2020) have embraced an auto-encoder approach, utilizing GNNs to encode node representations and decode edges based on relationships between node pairs. Concurrently, alternative frameworks (Zhang & Chen, 2018; Teru et al., 2020) explicitly encode the subgraph surrounding each node pair for enhanced link prediction accuracy. While these methods have demonstrated superior performance to traditional auto-encoder approaches (Zhang et al., 2020) and offer inductive learning capabilities, their requirement to materialize subgraphs for each link hinders scalability, especially with large graphs. Subsequent advancements (Zhu et al., 2021b; 2023; Zhang & Yao, 2022; Zhang et al., 2023a; Liu et al., 2024) have focused on learning path-based representations for pairs, with innovative labeling strategies yielding state-of-the-art performance.

**Text-based Methods.** Text-based methods utilize pre-trained language models (Devlin et al., 2019; Liu et al., 2019; Lan et al., 2020; Dong et al., 2019; Bao et al., 2020) to derive text representations of nodes and relations, upon which reasoning models are developed. KG-BERT (Yao et al., 2019) applies BERT for knowledge graph reasoning, treating triples as textual sequences to calculate a scoring function effectively for entities and relations by inputting their descriptions. StAR (Wang et al., 2021a) combines textual encoding and graph embedding techniques for heterogeneous graph reasoning, utilizing a siamese-style textual encoder for knowledge graph triple processing and incorporating structure learning for enhanced efficiency and performance. HittER (Chen et al., 2021) introduces a hierarchical Transformer model that learns entity and relation representations by considering local graph neighborhoods. It comprises two Transformer blocks for capturing entity-relation interactions and aggregating relational context, with a masked entity prediction task for balancing contextual information and original entity features. SimKGC (Wang et al., 2022) proposes an efficient contrastive learning technique using pre-trained language models for knowledge graph completion, employing various types of negative samples and the InfoNCE loss function to enhance performance. In contrast, KGT5 (Saxena et al., 2022) adopts a sequence-to-sequence approach, treating the graph reasoning task as a text generation problem to streamline the model architecture, achieving state-of-the-art results with a notably reduced model size.

## 5 CONCLUSION

This paper presents CoST, an optimized framework designed for graph reasoning tasks. While existing methods have demonstrated remarkable performance, the integration of structural and textual data through simultaneous training of graph neural networks and language models poses notable challenges, especially with respect to the complexity of real-world graph structures. To address these challenges related to training on extensive graph datasets, we employ an alternating training strategy for the graph neural network (GNN) model and pre-trained language model (PLM) with a variational objective. Empirical assessments validate the efficacy of our proposed model.

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

## A  DISCUSSION AND COMPARISON

In the recent research community, a subset of researchers have studied how to combine structural information and textual information. The subsequent paragraphs delineate our motivation and distinguish our work from existing research works.

**Focusing on Graph Reasoning Task.**   In this paper, we focus on the graph reasoning task, while most previous studies (Zhu et al., 2021a; Yang et al., 2021; Zhao et al., 2023; Yan et al., 2023) attempt to exploit both textual information and graph structures to address node classification tasks. Contrasting with node-level tasks like node classification, graph reasoning necessitates the model to make predictions based on pairwise representations, presenting a substantial challenge in designing GNN-based and PLM-based architectures, as well as in handling the interaction between these two models. Moreover, the graph reasoning task necessitates predicting across the entire node set of a specific graph, presenting a scalability challenge in contrast to classification tasks with predefined categories.

**Learning Effective Embedding.**   There are also partial works that try to combine textual information and structural information to solve reasoning tasks such as knowledge graph completion (He et al., 2024a). However their emphasis lies in enhancing the performance of two isolated models, particularly in sparse graph scenarios, our model goes beyond solely improving performance to encompass learning effective text representation. By employing text representations as a connective layer between GNN-based models and PLM-based models, our approach enhances the optimization of diverse models and enhances the efficacy of text representations.

## B  PROOF OF THEOREM 2.1

*Proof.*

$$
\log p_\phi \left( O_{(h,r)} | h, r, \mathcal{E}_{/\{(h,r,\hat{t})\}_{\hat{t} \in O_{(h,r)}}}, \mathcal{T} \right)
$$

$$
= \log \frac{p_\phi \left( O_{(h,r)}, H_{(h,r)} | h, r, \mathcal{E}_{/\{(h,r,\hat{t})\}_{\hat{t} \in O_{(h,r)}}}, \mathcal{T} \right)}{p_\phi \left( H_{(h,r)} | h, r, \mathcal{E}, \mathcal{T} \right)}
$$

$$
= \log \frac{p_\phi \left( O_{(h,r)}, H_{(h,r)} | h, r, \mathcal{E}_{/\{(h,r,\hat{t})\}_{\hat{t} \in O_{(h,r)}}}, \mathcal{T} \right) q_\theta \left( H_{(h,r)} | h, r, \mathcal{T} \right)}{p_\phi \left( H_{(h,r)} | h, r, \mathcal{E}, \mathcal{T} \right) q_\theta \left( H_{(h,r)} | h, r, \mathcal{T} \right)}
$$

$$
= \log \frac{p_\phi \left( O_{(h,r)}, H_{(h,r)} | h, r, \mathcal{E}_{/\{(h,r,\hat{t})\}_{\hat{t} \in O_{(h,r)}}}, \mathcal{T} \right)}{q_\theta \left( H_{(h,r)} | h, r, \mathcal{T} \right)} - \log \frac{p_\phi \left( H_{(h,r)} | h, r, \mathcal{E}, \mathcal{T} \right)}{q_\theta \left( H_{(h,r)} | h, r, \mathcal{T} \right)} \qquad (13)
$$

$$
= \int_{q_\theta \left( H_{(h,r)} | h, r, \mathcal{T} \right)} \log \frac{p_\phi \left( O_{(h,r)}, H_{(h,r)} | h, r, \mathcal{E}_{/\{(h,r,\hat{t})\}_{\hat{t} \in O_{(h,r)}}}, \mathcal{T} \right)}{q_\theta \left( H_{(h,r)} | h, r, \mathcal{T} \right)}
$$

$$
- \int_{q_\theta \left( H_{(h,r)} | h, r, \mathcal{T} \right)} \log \frac{p_\phi \left( H_{(h,r)} | h, r, \mathcal{E}, \mathcal{T} \right)}{q_\theta \left( H_{(h,r)} | h, r, \mathcal{T} \right)}
$$

$$
= \mathbb{E}_{q_\theta \left( H_{(h,r)} | h, r, \mathcal{E}, \mathcal{T} \right)} \left[ \log \frac{p_\phi \left( O_{(h,r)}, H_{(h,r)} | h, r, \mathcal{E}_{/\{(h,r,\hat{t})\}_{\hat{t} \in O_{(h,r)}}}, \mathcal{T} \right)}{q_\theta \left( H_{(h,r)} | h, r, \mathcal{T} \right)} \right] +
$$

$$
D_{\mathrm{KL}} \left( q_\theta \left( H_{(h,r)} | h, r, \mathcal{T} \right) || p_\phi \left( H_{(h,r)} | h, r, \mathcal{E}, \mathcal{T} \right) \right)
$$

Combining all queries, we obtain the final objective in Theorem 2.1. □

# C  MORE DETAILS OF EXPERIMENTS

## C.1  DATASETS

### C.1.1  HOMOGENEOUS GRAPH DATASETS

**AmazonSports & AmazonClothing.**    The two datasets consist of e-commerce networks, where each node represents a product available on Amazon, and an edge between two nodes signifies frequent co-purchases, which are created by He et al. (2024b) through randomly selecting 20,000 nodes from original datasets.

**MAGGeology & MAGMath.**    The two datasets are scholarly networks, with each node symbolizing a research paper, and an edge linking two nodes indicating one paper citing the other, which are created by He et al. (2024b) through randomly selecting 20,000 nodes from original datasets.

**CitationV8.**    CitationV8 is a directed graph dataset that illustrates the citation relationships among selected papers extracted from DBLP (Tang et al., 2008). Adhering to the parameters outlined in Yan et al. (2023), the experimental setup involves randomly excluding two references for each source paper. Subsequently, the prediction model aims to rank the two omitted references higher than 2,000 negative references and candidates.

**GoodReads.**    GoodReads is sourced from the world's largest book review platform. Nodes correspond to books, and edges are formed based on similarity relationships between books available on the website. Moreover, we adhere to the parameters set in Yan et al. (2023), which involve selecting 5,000 randomly sampled negative instances for evaluation purposes.

### C.1.2  HETEROGENEOUS GRAPH DATASETS

**FB15k237.**    FB15k237 is a knowledge graph reasoning dataset derived from FB15k, which originates from Freebase (Bollacker et al., 2008). While FB15k (Bordes et al., 2013) comprises $1,345$ relations, $14,951$ entities, and $592,213$ edges, numerous triples are reciprocal, leading to leakage in data splits between training, testing, and validation sets. Therefore, FB15k-237 was introduced by Toutanova & Chen (2015) to mitigate the issue of inverse relation test leakage in the evaluation datasets.

**WN18RR.**    WN18RR is a knowledge graph reasoning dataset derived from WN18, a subset of WordNet (Miller, 1994). WN18 comprises $18$ relations and $40,943$ entities. To prevent inverse relation test leakage arising from the inversion of triples in the training set, the WN18RR dataset was specifically formulated for evaluation purposes.

**Wikidata5M.**    Wikidata5M is a large-scale knowledge graph dataset accompanied by an aligned corpus, merging information from the Wikidata (Vrandecic & Krötzsch, 2014) and Wikipedia pages. Each entity within Wikidata5M is connected to a corresponding Wikipedia page, facilitating the assessment of link prediction for unfamiliar entities. Moreover, the dataset offers both transductive and inductive data partitions.

## C.2  IMPLEMENTATION DETAILS

### C.2.1  IMPLEMENTATION OF PLM MODEL

The implementation of the PLM model in CoST involves a pretrained language model and a decoder network. For the pretrained language model, we take minilm Wang et al. (2020; 2021b) as the example. The decoder computes the score for the candidate fact $(h', r', t')$ using an inner product between representation vectors as follows:

$$\text{Score}(h', r', t') = \langle g\left(\texttt{MiniLM}(h') * \texttt{MiniLM}(r')\right), \texttt{MiniLM}(t') \rangle, \tag{14}$$

where $g$ denotes a multi-layer perceptron (Rumelhart et al., 1987).

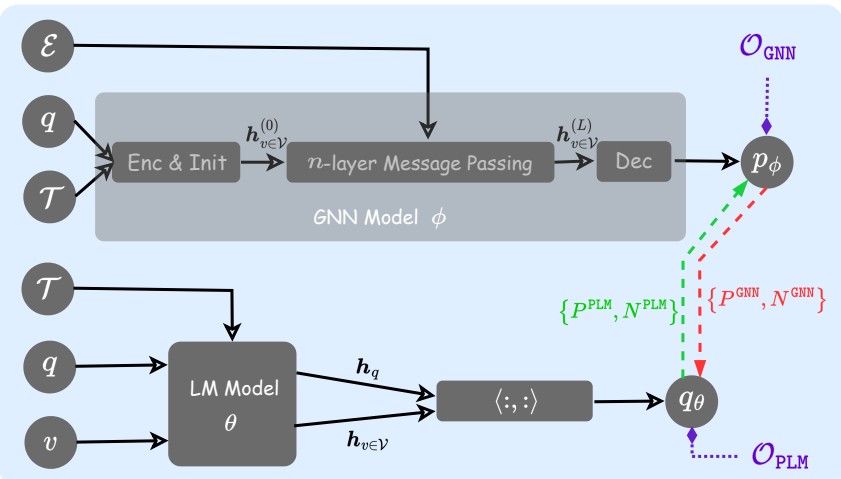

Figure 4: Detailed illustration of the overall optimization process of CoST.

### C.2.2 IMPLEMENTATION OF GNN MODEL

We implement the GNN model in CoST with a similar architecture to the model introduced in Zhu et al. (2023). We will introduce the detail implementation of ENC, INIT, MSG, AGG, UPD, and DEC.

**ENC.** The ENC function is utilized to encode textual information into continuous embeddings. In this paper, we utilize the PLM model denied before to implement ENC.

**INIT.** The INIT function serves to initialize representations for nodes and relations. In our implementation, node representation is initialized utilizing its textual embedding and a label vector:

$$\text{INIT}(u|h, r, \mathcal{T}) = g\left([\text{ENC}(\mathcal{T}_u) : \mathbf{1}_{u=h}]\right), \qquad (15)$$

where $[:]$ denotes a concatenation function and $g$ represents a multi-layer perceptron (Rumelhart et al., 1987). The label vector $\mathbf{1}_{u=h}$ comprises all one vector for the query node $h$ and all zero vectors for other nodes. Conversely, relation representation is only initialized by its textual embedding:

$$\text{INIT}(r) = \text{ENC}(\mathcal{T}_r) \qquad (16)$$

**MSG.** We employ the message function drawing inspiration from DistMult (Yang et al., 2015). To be more specific, the MSG function is implemented by multiplying node representation $\boldsymbol{h}_u$ by relation representation $\boldsymbol{e}_r$.

**AGG.** AGG function is implemented as the summation aggregation.

**UPD.** The UPD function is implemented using a neural network. Following the summation aggregation, the resultant representation is combined with the original node representation and fed into a multi-layer perceptron (Rumelhart et al., 1987) for updating the representation.

**DEC.** The DEC function is utilized to compute the score for each node. A multi-layer perceptron (Rumelhart et al., 1987) is employed, followed by a sigmoid function to calculate the score.

**Node and Edge Selection.** To enhance scalability, we apply a sampling mechanism to efficiently select important nodes and edges during message-passing computations Zhu et al. (2023); Zhang et al. (2023a) on larger graphs. During each iteration of message-passing, a neural network, identical in architecture and parameters to DEC, is utilized to determine significant scores for each node. The process involves selecting the top-$K$ crucial nodes and subsequently identifying the top-$L$ important edges based on the target node scores. Through this sampling approach, the GNN model can effectively handle large-scale graphs.

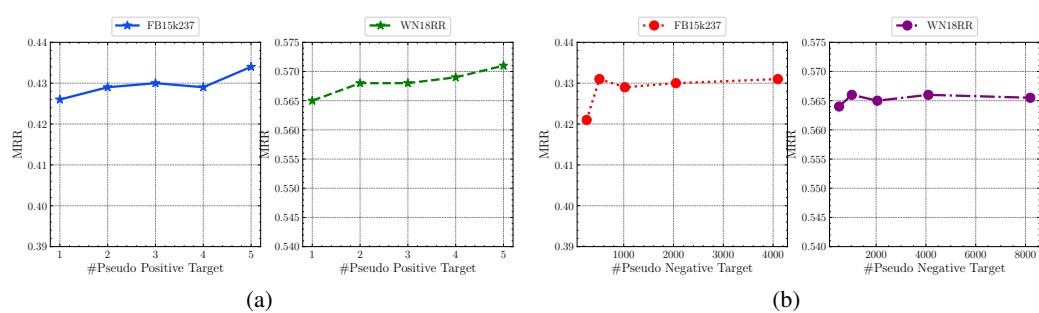

(a)        (b)

Figure 5: Illustration of performance w.r.t. the number of pseudo targets on the FB15k237 and WN18RR datasets.

### C.3 HYPERPARAMETERS

In our PLM model, we adjust the input token lengths to $\{64, 128, 256\}$ based on the textual information's varying lengths. During pretraining, we utilize a batch size of $512$, which is reduced to $64$ when alternately training, while setting the learning rate to $5e - 5$. For the GNN model, we fix the dimension size at $32$, the batch size at $64$, and the learning rate at $5e - 3$. The parameters such as the number of pseudo target samples are fine-tuned based on the performance of the validation set.

### C.4 HARDCORE CONFIGURATIONS

We conduct all experiments with:

- Operating System: Ubuntu 22.04.3 LTS.
- CPU: Intel (R) Xeon (R) Platinum 8358 CPU @ 2.60GHz with 1TB DDR4 of Memory and Intel Xeon Gold 6148 CPU @ 2.40GHz with 384GB DDR4 of Memory.
- GPU: NVIDIA Tesla A100 SMX4 with 80GB.
- Software: CUDA 12.1, Python 3.9.19, PyTorch (Paszke et al., 2019) 2.3.0.

## D ADDITIONAL EXPERIMENTS

**Performance w.r.t. Number of Pseudo Targets.** In the context of the `CoST` framework, we employ pseudo targets generation to train the model. To assess the correlation between the model's performance and the number of pseudo targets, we conduct pertinent experiments, the results of which are displayed in Figure 5. Demonstrating commendable resilience concerning the number of pseudo targets, our model exhibits notable performance enhancements even with a limited quantity of pseudo targets.

**Performance w.r.t. Different PLM Architectures.** Additionally, we utilize various pre-trained language models to assess the consistency of performance within the `CoST` framework, as depicted in Figure 6. Our model consistently demonstrates enhancements across three distinct PLMs, with stronger PLM performance correlating with amplified `CoST` performance.

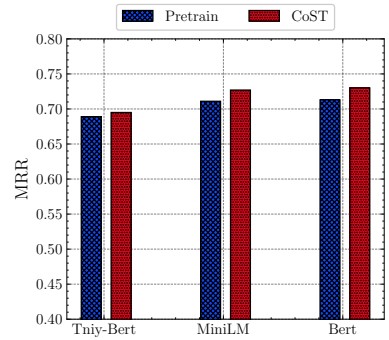

Figure 6: Results of `CoST` on CitationV8 dataset based on Tiny-Bert (Jiao et al., 2020), MiniLM (Wang et al., 2020), and Bert (Devlin et al., 2019).

