# OpenReview forum: "Combining Structure and Text: Learning Representations for Reasoning on Graphs"
_ICLR.cc/2025/Conference — Submitted to ICLR 2025_

### Official Review · Reviewer_jTir · 2024-10-27

**Soundness:** 3
**Presentation:** 3
**Contribution:** 3
**Rating:** 6
**Confidence:** 4

**Summary:**

In this paper, the authors propose a comprehensive framework that addresses the challenges associated with processing graph data by combining both structural attributes (the connections between nodes) and textual information (node labels or node descriptions). The paper aims to improve the performance of various graph-based reasoning tasks, including link prediction, node classification, and knowledge graph completion. These models are trained on large datasets to learn representations that can effectively distinguish between different nodes based on their structural position within the graph and their associated text features.

**Strengths:**

1. The paper is well-structured, making it accessible to both researchers and practitioners in the field. It provides clear explanations for complex concepts and algorithms, supported by illustrative examples that aid understanding.

2. The clarity of the paper is good. The authors use appropriate terminology throughout the document, making it accessible to researchers with a background in machine learning and graph theory.

**Weaknesses:**

1.  It would be beneficial to provide a detailed description of the experimental setup, including data preprocessing steps, model configurations, hyper-parameters tuning process, and any specific challenges encountered during implementation.

2. The authors could include a more detailed discussion on model evaluation metrics that specifically address the integration of structure and text.

**Questions:**

How much do hyperparameters affect the method proposed in this work?

---

> ### Author Response · Authors · 2024-11-17
> **Response to Reviewer jTir Part I**
>
> Thank you for your positive comments on our work! Below are our responses to your concerns.
>
> > More details about experiments and evaluation metrics
>
> Thanks for your valuable suggestions. We provide details about datasets, hyperparameters, and implementation details in the Appendix, and we will add more necessary details in the later version. For the evaluation metrics, we choose the widely used metrics as described in Line 366.
>
> > Impact of hyperparameters
>
> Our approach is based on existing LM and GNN methods, the effects of hyperparameters have been explored in their original papers, and we simply choose the optimal combination of parameters. In addition, in Appendix D we discuss the effect of the newly key hyperparameter introduced by our method, i.e. the number of pseudo-targets, and we can observe that our method is relatively stable for the performance of this parameter.
>
> ---
>
> Finally, thank you again for your efforts and we will refer to your comments based on improving our paper.

---

> > ### Author Response · Authors · 2024-11-26
> > **Kindly Reminder**
> >
> > Thanks for your reviews and recognition. We hope our response can be a good answer to your concerns. If your have other questions, please let us know.

---

> ### Author Response · Authors · 2024-12-02
>
> Dear reviewer jTir,
>
> We appreciate your reviews. As the discussion period ends very soon, may we ask if our response to your concerns is helpful? We are always happy to answer any questions and improve our work.
>
> Sincerely,
>
> Authors of paper3161

---

### Official Review · Reviewer_LMJF · 2024-11-03

**Soundness:** 3
**Presentation:** 2
**Contribution:** 2
**Rating:** 5
**Confidence:** 4

**Summary:**

This paper aims to combine structure and text information for solving reasoning tasks on graphs. The authors propose CoST, a framework that employs graph neural networks (GNNs) to capture structural features and pretrained language models (PLMs) to capture textual features respectively. Since it is challenging to jointly optimize the GNN and PLM, the authors leverage the variational lower bound of the joint objective and optimize the GNN and PLM in a cyclical manner. The authors demonstrate the effectiveness of CoST on 9 datasets, including both homogeneous and heterogeneous graphs. CoST outperforms both GNN-based and LLM-based methods that only uses information from a single modality.

**Strengths:**

- This paper studies the problem of combining structure and text, which is a general and important problem in the community of graph representation learning.
- The proposed CoST has clear motivations from variational Bayesian perspective.
- CoST provides robust performance gain over a wide range of datasets and several base models. The empirical contribution is solid and likely to be applicable to other models and datasets.

**Weaknesses:**

- The position and the contribution of this paper is not very clear. Combining structure and text has been widely studied in the context of text-attributed graphs (TAGs) [1, 2], and I don’t think the setting of this paper is a contribution given its similarity to TAGs. Regarding the proposed CoST, several papers have used the same idea of variational EM algorithm to optimize two models capturing two views of graphs [3, 4, 5]. There is scarcely any novelty for the proposed method.
- It is not very fair to compare CoST against GNN-based and LLM-based methods that only use features from one modality. Based on this comparison, we can only conclude that combining two modalities is effective, which is a known lesson in many machine learning domains. We can’t conclude that CoST is an advanced method for combining two modalities. The authors should compare CoST against some baselines (e.g. [1]) that fuse structural and textual information for graph reasoning.
- The abstract and introduction are not well written. The authors focused too much on the setting of combining structure and text, which is important but not a big challenge in the community. I recommend the authors to rethink the challenges in this setting and discuss more about how their method technically solves the challenges. The authors may also remove Figure 1, as it is not related to the challenges.

[1] Harnessing Explanations: LLM-to-LM Interpreter for Enhanced Text-Attributed Graph Representation Learning. He et al. ICLR 2024.

[2] One for All: Towards Training One Graph Model for All Classification Tasks. Liu et al. ICLR 2024.

[3] Efficient Probabilistic Logic Reasoning with Graph Neural Networks. Zhang et al. ICLR 2020.

[4] RNNLogic: Learning Logic Rules for Reasoning on Knowledge Graphs. ICLR 2021.

[5] Learning on Large-scale Text-attributed Graphs via Variational Inference. Zhao et al. ICLR 2023.

**Questions:**

- In Equation 3, how do you define the probability over an answer set $O_{(h, r)}$? Equation 1 & 2 only define the probability over a single answer.
- How to understand the meaning of the latent variable $H_{(h, r)}$? It looks like $V_{/O(h, r)}$ excludes the ground truth answers from all entities. How can you call it the candidate target node set for the query (h, r, ?)?
- Equation 10 should be log probabilities, not probabilities.
- Why do you omit $H_{(h, r)}$ in $p_\phi$ in the second line of Equation 10?
- Line 461 - 463: I understand CoST pretrained GNN only uses structural information. How do language models come up here?
- $t$ in Equation 12 doesn’t need to be bold.
- It’s a little bit complicated for CoST to use two optimization stages. Is there any reason for not using a mixed objective of pseudo labels and ground truth like in [5]?

---

> ### Author Response · Authors · 2024-11-17
> **Response to Reviewer LMJF Part I**
>
> Thank you for your efforts and detailed review. We are glad that you recognize the motivation and effectiveness of our work. In response to some of your concerns and questions, we make the following responses.
>
> > Clarification of position and the contribution
>
> 1. **Clarification of task**
>
> Due to the unique characteristics of text-attributed graphs, several studies have explored the inference of representation learning and downstream tasks by integrating text attributes and structural information. However, most of this work (as works [1][2][5] you have given) has primarily focused on:
> * Node-level and graph-level classification tasks.
> * Homogeneous datasets.
>
> In contrast, this paper works on the graph reasoning task, leading to the following key differences:
> *  Graph reasoning task requires inferencing a set of answering nodes based on graph data for a given query node (and possible query relation). Unlike node-level and graph-level classification tasks, where the classification label is fixed, the graph reasoning task requires considering different query conditions and predicting dynamic answer node sets.
> * Compared with the link-level classification task, the definition of graph reasoning task is more general, which requires the reasoning result of any candidate node, while the link-level classification problem always makes inference between two given nodes. On a graph with $ n $ nodes, The graph reasoning model usually obtains the reasoning result of any candidate node through one-step forward, while the link-level classification model requires $ n $ times forward inference.
> * Additionally, we consider heterogeneous data, which requires additional modeling for relational dimensions.
>
> 2. **Clarification of CoST**
>
> Although our paper and some other work (as works [3][4][5] you have given) also use the idea of variational EM, the specific challenges and differences of our work differentiate it from the aforementioned work. For example:
> * Due to the dynamic set of answer nodes, the estimation of the posterior distribution is not as straightforward as the classification problem of fixed label sets;
> * Due to the existence of query conditions and the requirement of pairwise representation, there are specific challenges in the construction of LM and GNN models and the interaction between them.
>
> Also, we have also discussed similar points in Appendix A of the paper, which you can refer to.
>
> > About comparisons with GNN-based and LLM-based baselines
>
> First, we want to clarify that the naming of some baselines as GNN-based or LM-based does not imply that they solely utilize structural information or text information. For example, GNN-based baselines (e.g., those listed in Table 2 and Table 3) use fixed text embeddings as features, while LM-based baselines (e.g., Topological Contrastive Learning based Method) incorporate structural information through structural training methods. For scalability reasons, these baselines often fix certain models, which limits their performance. This limitation is a primary motivation for our work. Therefore, we believe it is reasonable to compare our approach with these baselines.
>
> Second, we have made efforts to include more GNN+LLM baselines. However, these methods either: 1) Are not suitable for graph reasoning tasks [1][5], or 2) Require very high computational overhead when run on graph inference tasks [2]. Nevertheless, we have done our best to incorporate some GNN+LLM baselines (e.g., LinkGPT, LLaGA) on small-scale benchmarks.
>
> > About the writing of abstract and introduction
>
> Thank you for your suggestions. Previous work in graph reasoning has focused less on the text modality, which is why we have emphasized this part extensively. However, your suggestions are very valuable, and we will modify our manuscript accordingly to highlight the specific challenges and contributions of our work. We appreciate your feedback.
>
> ---
>
> Please see Part II

---

> ### Author Response · Authors · 2024-11-17
> **Response to Reviewer LMJF Part II**
>
> > Q1: In Equation 3, how do you define the probability over an answer set $ O_{(h,r)} $? Equation 1 & 2 only define the probability over a single answer.
>
> Similar to Equation 5, the distribution is defined by $ p _ {\phi}\left( O_{(h,r)} \vert h, r, \mathcal{E} _ {/\{(h,r,\hat{t})\} _ {\hat{t} \in O _ {(h,r)}}}, \mathcal{T} \right) = \prod _ {\hat{t}\in O _ {(h,r)}} p _ {\phi}\left( \hat{t} \vert h, r, \mathcal{E} _ {/\{(h,r,\hat{t})\} _ {\hat{t} \in O _ {(h,r)}}}, \mathcal{T} \right) $.
>
> > Q2: How to understand the meaning of the latent variable $ H _ {(h,r)} $? It looks like $ \mathcal{V} _ {/O _ {(h,r)}} $excludes the ground truth answers from all entities. How can you call it the candidate target node set for the query $ (h,r,?) $?
>
> Most real-world graph datasets are incomplete, so the $ O _ {(h,r)} $is only the observed answer node set. $ H _ {(h,r)} = \mathcal{V} _ {/O _ {(h,r)}} $could contain some unobserved answer nodes, so we call it candidate target node set.
>
> > Q3:  Equation 10 should be log probabilities, not probabilities.
>
> Thank you for pointing out the problem, we will fix it.
>
> > Q4:  Why do you omit $ H_{(h,r)} $in $ p_{\phi} $ in the second line of Equation 10?
>
> We have $ O_{(h,r)} \cup H_{(h,r)} = \mathcal{V} $, so we can use the summation over $ t \in \mathcal{V} $.
>
> > Q5: Line 461 - 463: I understand CoST pretrained GNN only uses structural information. How do language models come up here?
> >
>
> We leverage the pretrained-language model to generate text embeddings and the generated embeddings is the input to the GNN model.
>
> > Q6: It's a little bit complicated for CoST to use two optimization stages. Is there any reason for not using a mixed objective of pseudo labels and ground truth.
>
> As we mentioned in Line 143-145, to jointly optimizing LM and GNN is not tractable, especially for large-scale datasets. So in this work, we leverage an alternative optimization process. [5] also includes two-stage optimization as described in Section 4.3 and 4.4 of the original paper.
>
> ---
>
> Finally,  thank you again for your careful review, your comments are very helpful to improve our paper. I hope our response has solved your questions, and if you have any questions, please don't hesitate to let us know.

---

> > ### Author Response · Authors · 2024-11-26
> > **Kindly Reminder**
> >
> > We are very grateful for your review and suggestions on our paper, which are very important to improve our work. At the same time, we kindly ask you to check our response. We will be very grateful if our response can address your concerns to improve your rating accordingly. If you have any further questions, please do not hesitate to let us know.

---

> ### Comment · Reviewer_LMJF · 2024-11-26
>
> Thanks the authors for their proper response.
>
> I agree with the authors' clarification of the task. With that being said, given the well-known nature of combining two modalities with variational inference, I can hardly see any new insight or lesson from this paper. Hence, the contribution of this paper seems limited to applying an existing methodology to a new task.
>
> For the experiments, the authors have convinced me that the baselines for homogeneous datasets are strong enough. However, my concern remains regarding the baselines for heterogeneous datasets, as none of them consider both structure and text.
>
> Based on these points, I decide to keep my rating.

---

> > ### Author Response · Authors · 2024-11-27
> > **Followup Response to Reviwer LMJF**
> >
> > Thanks for your response, regarding your remaining concerns, we would like to provide the following additional responses.
> >
> > > C1: Concern about `the contribution of this paper seems limited to applying an existing methodology to a new task`.
> >
> > Perhaps we can think in terms of the EM algorithm, which is a classic idea that has been applied to many different works and scenarios. It needs to model different types of latent variables when applying EM in different scenarios. In this paper, the learning of latent variables can be abstracted into a **dynamic set prediction (node-set) task**, so we believe that we provide insights for the set latent variables learning problems. In contrast, most of the previous works model a single item (node).
> >
> > > C2: Concern remains regarding the baselines for heterogeneous datasets, as none of them consider both structure and text
> >
> > We indeed understand your concern about the experimental baselines for reasoning on heterogeneous graphs, but as we mentioned in the previous response, structure+text baselines on heterogeneous graphs are rare and usually perform poor.
> > Nevertheless, we try to report the StAR [1] and as CSProm-KG structure-enhanced LM-based baselines for your reference, which **leverage structure-augmented text representation for reasoning** and we also report the performance of NBFNet and AStarNet, where we **modify original frameworks by leveraging text embedding generating from MiniLM (aligned with CoST) as inputs**.
> > | | FB15k-237-mrr | WN18RR-mrr |
> > | -- | --| -- |
> > |StAR$_{bert-base}$	|36.5	|55.1|
> > |CSProm-KG$_{bert-large}$ |35.8	|**57.5**|
> > |NBFNet$_{text}$	|41.4	|55.3|
> > |A$^*$Net$_{text}$|	40.9	|55.7|
> > |CoST |**42.7**	|**56.8**|
> >
> > We can observe that CoST outperforms these baselines.  LM-based baselines are **hard to utilize structural information** due to the characteristics of the language model even with structural information enhancement.  For modified NBFNet and A$^*$Net, we cannot observe significant performance improvement and even observe a slight performance degradation, we attribute it to **the gap between GNN desired input space and the fixed text embeddings and fixed text embeddings will impair GNN's capabilities** (compared with learnable embedding in the original NBFNet and A*Net).  This highlights the effectiveness of CoST, which jointly trains GNN and LM for reasoning on graphs.
> >
> > [1] Structure-Augmented Text Representation Learning for Efficient Knowledge Graph Completion. in WWW 2021.
> >
> > [2] Dipping PLMs Sauce: Bridging Structure and Text for Effective Knowledge Graph Completion via Conditional Soft Prompting. in ACL 2023.
> >
> > ---
> > We hope our responses will be helpful.  Should you have other questions, please let us know. We looking forward to your reply.

---

> > > ### Author Response · Authors · 2024-11-28
> > > **Additional Experimental Results**
> > >
> > > We also want to additionally report the results for RED-GNN and AdaProp with text embedding as inputs for your reference.
> > > | | FB15k-237-mrr | WN18RR-mrr |
> > > | -- | --| -- |
> > > |StAR$_{bert-base}$	|36.5	|55.1|
> > > |CSProm-KG$_{bert-large}$ |35.8	|**57.5**|
> > > |NBFNet$_{text}$	|41.4	|55.3|
> > > |A$^*$Net$_{text}$|	40.9	|55.7|
> > > |RED-GNN$_{text}$	|38.3	|54.8|
> > > |AdaProp$_{text}$|	41.2	|55.3|
> > > |CoST |**42.7**	|**56.8**|
> > > The experimental results can further verify our hypothesis in the previous response.

---

> ### Author Response · Authors · 2024-12-02
>
> Dear reviewer LMJF,
>
> We appreciate your reviews. As the discussion period ends very soon, may we ask if our response to your follow-up concerns is helpful? If the concerns have been fully addressed, we would be grateful if you could consider updating your score accordingly. Thank you again for your thoughtful consideration and hard work.
>
> Sincerely,
>
> Authors of paper3161

---

### Official Review · Reviewer_99Gn · 2024-11-04

**Soundness:** 3
**Presentation:** 4
**Contribution:** 2
**Rating:** 5
**Confidence:** 3

**Summary:**

This article proposes a new optimization method tailored for the combination of GNNs and PLMs. This work is motivated by an ICLR 2023 paer titled "GLEM - Learning on Large-scale Text-attributed Graphs via Variational Inference in ICLR 2023".
In this article, the authors conduct multiple sets of experiments on graph datasets from various domains, with the experimental conclusions validating the effectiveness of their new method in graph search-related problems.This new optimization method outperforms traditional graph structure (Struct)-based methods, PLM-based methods, and some mainstream methods combining GNNs and PLMs across four mainstream models.

**Strengths:**

This article is well written with clear presentation of the new optimization method, and their experiments are rigorous  with repeatability.

**Weaknesses:**

Overall, this paper may represent a significant algorithmic contribution, but it requires supplementing some theoretical details in the article:
- In the proof of Theorem 2.1 (Equation 13), the derivation of the last equality lacks detail. It requires a detailed elaboration on how the prefix $\mathbf{E}$ is added to the first term and why the second term lacks a $\mathcal{E}$ compared to the description in Theorem 2.1. Could you provide a step-by-step derivation of the last equality, specifically explaining the addition of the expectation operator to the first term and the absence of it in the second term? This would help clarify the mathematical reasoning behind Theorem 2.1.

**Questions:**

1. Supplementary figures are needed. Could you could include a flowchart or diagram that shows the step-by-step process of how a query is processed through the PLM and GNN components of the CoST framework? This would enhance the reader's understanding of the model's architecture and operation.
2. In Table 2 and Table 3, $H@1$ is only computed for the AmazonSports, AmazonClothing, MAGGeology, and MAGMath datasets, while $H@10$, $H@50$, $H@100$ are only computed for the CitationV8 and GoodReads datasets. Could you explain the rationale behind choosing these specific metrics for each dataset, and whether this choice affects the comparability of results across datasets?

---

> ### Author Response · Authors · 2024-11-17
> **Response to Reviewer 99Gn Part I**
>
> Thank you for your constructive review and suggestions! We are pleased that you recognize our contribution to the work, experimentation, and writing. Below, we will respond to your concerns.
>
> > In the proof of Theorem 2.1 (Equation 13), the derivation of the last equality lacks detail. It requires a detailed elaboration on how the prefix $ \mathcal{E} $ is added to the first term and why the second term lacks a $ \mathcal{E} $ compared to the description in Theorem 2.1. Could you provide a step-by-step derivation of the last equality, specifically explaining the addition of the expectation operator to the first term and the absence of it in the second term? This would help clarify the mathematical reasoning behind Theorem 2.1.
>
> **1.how the prefix **$ \mathbb{E} $** is added to the first term**
>
> From the previous step of the last step, we have:
>
> $$
> \log p _ {\phi} \left( {O _ {(h,r)}} | h, r, \mathcal{E} _ {/\{(h,r,\hat{t})\} _ {\hat{t}\in O_{(h,r)}}}, \mathcal{T} \right) = \log \frac{p _ {\phi} \left( {O _ {(h,r)}}, {H _ {(h,r)}} | h, r, \mathcal{E} _ {/\{(h,r,\hat{t})\} _ {\hat{t}\in O _ {(h,r)}}}, \mathcal{T} \right)}{q _ {\theta}\left( {H _ {(h,r)}} | h, r, \mathcal{E}, \mathcal{T} \right)} - \log \frac{p _ {\phi} \left( {H _ {(h,r)}} | h, r, \mathcal{E}, \mathcal{T} \right)}{ q _ {\theta}\left( {H _ {(h,r)}} | h, r, \mathcal{E}, \mathcal{T} \right)}
> $$
>
> And then we integrate $ q_{\theta}\left( {H_{(h,r)}} | h, r, \mathcal{E}, \mathcal{T} \right) $ on both sides of the equation. The formula on the left is independent of $ q _ {\theta}\left( {H_{(h,r)}} | h, r, \mathcal{E}, \mathcal{T} \right) $, so it remains the same, the first term $ \int _ {q _ {\theta}\left( {H _ {(h,r)}} | h, r, \mathcal{E}, \mathcal{T} \right)}\log \frac{p _ {\phi} \left( {O _ {(h,r)}}, {H _ {(h,r)}} | h, r, \mathcal{E} _ {/\{(h,r,\hat{t})\} _ {\hat{t}\in O_{(h,r)}}}, \mathcal{T} \right)}{q _ {\theta}\left( {H_{(h,r)}} | h, r, \mathcal{E}, \mathcal{T} \right)} $ in the formula on the right can be regarded as the expectation of $ \log \frac{p _ {\phi} \left( {O _ {(h,r)}}, {H _ {(h,r)}} | h, r, \mathcal{E} _ {/\{(h,r,\hat{t})\} _ {\hat{t}\in O _ {(h,r)}}}, \mathcal{T} \right)}{q _ {\theta}\left( {H _ {(h,r)}} | h, r, \mathcal{E}, \mathcal{T} \right)} $ with probability $ q _ {\theta}\left( {H _ {(h,r)}} | h, r, \mathcal{E}, \mathcal{T} \right) $, and the second term $ -\int _ {q _ {\theta}\left( {H _ {(h,r)}} | h, r, \mathcal{E}, \mathcal{T} \right)} \log \frac{p _ {\phi} \left( {H _ {(h,r)}} | h, r, \mathcal{E}, \mathcal{T} \right)}{ q _ {\theta}\left( {H _ {(h,r)}} | h, r, \mathcal{E}, \mathcal{T} \right)}  $ is the divergence of KL between $ p _ {\phi} \left( {H _ {(h,r)}} | h, r, \mathcal{E}, \mathcal{T} \right) $ and $ q _ {\theta}\left( {H _ {(h,r)}} | h, r, \mathcal{E}, \mathcal{T} \right) $. So we can draw conclusions to the last step.
>
> **2.why the second term lacks a **$ \mathcal{E} $****
>
> As mentioned in the paper (Line 153), $ q_{\theta}\left( {H_{(h,r)}} | h, r, \mathcal{E}, \mathcal{T} \right) $is modeled by LM model and the input to LM model is only related to $ h,r $and $ \mathcal{T} $, so actually we have$ q_{\theta}\left( {H_{(h,r)}} | h, r, \mathcal{E}, \mathcal{T} \right) = q_{\theta}\left( {H_{(h,r)}} | h, r, \mathcal{E}, \right) $. Sorry for the confusion, we will modify it in the later version to keep the description consistent.
>
> > Could you explain the rationale behind choosing these specific metrics for each dataset, and whether this choice affects the comparability of results across datasets?
>
> We chose the metrics following the works that present certain datasets. In our view, different metrics are employed for different datasets due to their distinct characteristics and purposes. A smaller value of $ n $ (e.g., $ H@1 $) indicates a greater emphasis on the precision of the model's output, which is more relevant for datasets like question-answering datasets. Conversely, a larger value of $ n
>  $ (e.g., $ H@50, H@100 $) indicates a greater focus on the recall performance of the model, which is more pertinent for datasets like recommendation datasets. We believe that this approach does not compromise the evaluation of different models' performance, as the overall performance of the same model generally exhibits a linear relationship with $ n $.
>
> > Supplementary figures
>
> Thanks for your suggestion, we have tried to add more necessary figures in the latest version.
>
> ---
> Finally, thank you again for your review. Your comments are very important for the improvement of our work. If you have other concerns, please let us know.

---

> > ### Author Response · Authors · 2024-11-26
> > **Kindly Reminder**
> >
> > We deeply value the constructive feedback you provided, which has been helpful in enhancing the quality of our manuscript. It is our hope that the revisions and responses have basically addressed your concerns.
> >
> > Should you find that our clarifications resolve your concerns, we would be grateful if you could reconsider your assessment score. We also welcome any additional questions or suggestions you may have.

---

> > > ### Author Response · Authors · 2024-12-02
> > >
> > > Dear reviewer 99Gn,
> > >
> > > We appreciate your reviews. As the discussion period ends very soon, may we ask if our response to your concerns is helpful? And If the concerns have been fully addressed, we would be grateful if you could consider updating your score accordingly.  ]
> > >
> > > Sincerely,
> > >
> > > Authors of paper3161

---

### Official Review · Reviewer_whTb · 2024-11-04

**Soundness:** 2
**Presentation:** 3
**Contribution:** 2
**Rating:** 6
**Confidence:** 3

**Summary:**

The manuscript introduces CoST (Combining Structure and Text), an advanced framework for graph reasoning that synergistically integrates structural information from Graph Neural Networks (GNNs) with textual data from Pre-trained Language Models (PLMs). The framework employs a novel alternating training mechanism designed to enhance the reasoning capabilities of both GNNs and PLMs when applied to real-world graph datasets. CoST effectively addresses scalability issues and optimizes the learning process by incorporating a variational objective, thus enriching both structural and contextual representations of graph entities. Empirical evaluations demonstrate that CoST achieves state-of-the-art performance across several benchmark datasets, including both homogeneous and heterogeneous graph types.

**Strengths:**

1. The primary strength of this paper lies in its alternating training framework, which iteratively optimizes both GNNs and PLMs. Unlike conventional approaches that either jointly train both models or concatenate their outputs, CoST introduces a two-stage optimization strategy that allows each model to refine its representations based on the complementary output of the other.

2. The authors conduct extensive experiments on multiple graph reasoning benchmarks. CoST consistently outperforms state-of-the-art baselines. The empirical results show significant improvements, validating the model's effectiveness across diverse datasets and various graph topologies.

**Weaknesses:**

1. Limited LLM Backbone Evaluation: The paper only utilizes MiniLM as the LLM backbone for the PLM component. To demonstrate the broader applicability of the proposed framework, additional experiments using prominent LLMs, such as llama, should be conducted. This would help ascertain whether the alternating training approach and the integration mechanism are generalizable across different LLM architectures.

2. Complexity of Training Paradigm: The alternating training paradigm, while effective, introduces considerable complexity in both model implementation and hyperparameter tuning. The generation of pseudo-targets by GNNs, coupled with the alternating nature of updates between GNN and PLM, results in multiple hyperparameters, complicating reproducibility.

3. Questions Regarding Pseudo-Target Method: The pseudo-target method is central to the alternating training paradigm in CoST, yet certain aspects remain under-explored. Specifically, why are particular pseudo-targets selected, and how does their quality influence the overall model performance? Are there explicit criteria guiding the selection of pseudo-targets, and how is their quality evaluated to ensure they are beneficial for PLM training? Clarifying these points would improve the understanding of how pseudo-targets contribute to model robustness and effectiveness.

**Questions:**

1. Why was MiniLM chosen as the LLM backbone? Could you experiment with other widely-used LLMs, such as llama, to demonstrate the generalizability of the approach?

2. Further Explanation of Pseudo Targets: The concept of using pseudo-targets from GNN predictions is innovative, yet it remains unclear how sensitive the model's performance is to the number of pseudo-targets chosen. Would varying the number of pseudo-targets significantly affect the model's performance? What criteria are used to select these pseudo-targets, and how is their quality assessed during training? Additional analysis on these questions could provide deeper insights into the robustness of the pseudo-target strategy.

3. Can you include a comparison with similar GNN+LLM works such as LinkGPT on additional benchmarks, as shown in Tables 3 and 4?

---

> ### Author Response · Authors · 2024-11-17
> **Response to Reviewer whTb Part I**
>
> Thanks for your time and valuable reviews. We are pleased with your recognition of the methods and effectiveness of our work, and we will address your main concerns below.
> > For various LM backbone and why not use LLM backbone
>
> * In our work, the role of the Language Model (LM) is to transform the textual information of nodes and edges into high-quality text embeddings. Therefore, we selected the MiniLM model, which is widely recognized for its effectiveness in text representation tasks and is one of the most downloaded models in the `sentence-transformers` library. Additionally, in Appendix D, we present the performance of our methods under different LM architectures, including Tiny-Bert, Bert, and MiniLM. Our results demonstrate consistent performance improvements across these architectures. So we believe that CoST can migrate well to other models.
> * The primary reason we did not conduct experiments with Large Language Models (LLMs) like LLaMA is that LLMs are typically designed with a decoder-only architecture, which excels in natural language generation but is not well-suited for generating text embeddings. In contrast, most contemporary text embedding models are based on encoder architectures, which are better suited for this task. However, we also provide some results based on llama3.2-1b for reference, although it has more parameters, it does not bring much performance improvement.
>
> |  | AmazonSports | AmazonClothing |
> | --- | --- | --- |
> |  | MRR | MRR |
> | LINKGPT | 87.1 | 90.2 |
> | CoST (paper) | 88.3 | 91.4 |
> | CoST (llama3.2-1b) | 87.9 | 89.3 |
>
>
> > Complexity of Training Paradigm
>
> * Our method can be seamlessly integrated into existing Language Model (LM) and Graph Neural Network (GNN) model codebases with minimal modifications.
> * As demonstrated in the experiments presented in Section 3.3 and Appendix D, our method exhibits rapid convergence when applied to pre-trained LM and GNN models, requiring fewer training iterations.
> * Furthermore, our method demonstrates robustness to the introduction of primarily additional hyperparameters, such as the number of pseudo examples.
>
> > Details about Pseudo Targets
>
> **1.Selection of pseudo-targets**
>
> As mentioned in Lines 232 and 266, we obtain pseudo targets by sampling from the distribution generated by the LM/GNN from the previous iteration, which can be viewed as a Monte Carlo estimation. In our method, this approach allows us to optimize the correlation between distributions, thereby minimizing the variational loss. Instead of using pseudo targets as direct input signals for teacher forcing training, the quality of the pseudo targets is less critical. In contrast, the number of pseudo targets is more important. This is because the number of pseudo targets influences the precision of the estimation.
>
> **2.Impact of the number of pseudo-targets**
>
> In Appendix D, we provide the experimental results regarding different numbers of pseudo-targets. It can be observed that as the number of pseudo-targets increases, the performance also improves, which is consistent with our previous hypothesis. However, this improvement is limited, and our model exhibits notable performance enhancements even with a relatively small number of pseudo-targets.
>
> ----
> Please see Part II

---

> ### Author Response · Authors · 2024-11-17
> **Response to Reviewer whTb Part II**
>
> > Experiments of GNN+LLM in Table 3 and Table 4
>
> Initially, we try to provide the performance of GNN+LLM baselines in all datasets. However, since the scale of datasets including CitationV8 and GoodReads is very large, the time overhead and GPU memory overhead of LLM-engaging methods are hardly acceptable. That's why many LLM-engaging methods will sample a small dataset for experiments. So we include the small-scale datasets including AmazonSports, AmazonClothing, MAGGeology, and MAGMath to compare with LLM-engaging methods. For the heterogeneous datasets FB15k237 and WN18RR, there lack baselines that combine the GNN and LLM, which mainly dues to the large scale of these datasets and the additional complexity of modeling the multi-relation. And most existing works [1][2] only work on link-level prediction, which is not scalable in our settings, since it needs $ n $ forward inference for reasoning over all $ n $ nodes. For example, on FB15k237 dataset, it will need 1w+ forward inferences for single query $ (h,r,?) $.  Based on this, we think our approach can provide insights on these datasets. There are indeed works that only use textual information to reason by LLM generation, however, because the structural information is essential in the graph reasoning task, this paradigm usually performs poorly, please refer to the following table for instance.
>
> | | WN18RR-H@1 |
> | --- | --- |
> | LLaMA-7B | 8.5 |
> | LLaMA-13B | 9.9 |
> | KG-LLaMA-7B [3] | 24.2 |
> | KG-LLaMA-13B [3] | 25.6 |
> | KG-LLaMA2-13B [3] | 26.9 |
> | KG-LLaMA2-13B + Struct [3] | 31.5 |
> | Ours | 51.0 |
>
>
> [1] LLaGA: Large Language and Graph Assistant. ICML 2024.
>
> [2] One for All: Towards Training One Graph Model for All Classification Tasks. ICLR 2024.
>
> [3] ExploringLargeLanguageModelsforKnowledgeGraphCompletion. Arxiv 2024.
>
> ----
> Lastly, thank you again for your careful review. We hope that our response will address your problems and if you still have additional problems, please don't hesitate to let us know.

---

> ### Comment · Reviewer_whTb · 2024-11-22
>
> Thank you for addressing my concerns! I will update my rating accordingly.

---

> > ### Author Response · Authors · 2024-11-25
> > **Thank you**
> >
> > Thanks again for your efforts and recognition.

---

### Author Response · Authors · 2024-11-17
**Common Response**

We express our gratitude to all reviewers for their time and patience in reviewing our paper. The strengths recognized by the reviewers include:

1. Focus on a general and important problem in graph learning [Reviewer LMJF]
2. Effective alternative training framework [Reviwer whTb] [Reviewer LMJF]
3. Extensive and rigorous experiments on multiple benchmarks [Reviwer whTb] [Reviewer 99Gn] [Reviewer LMJF]
4. Strong empirical results across diverse datasets and various graph topologies [Reviwer whTb] [Reviewer LMJF]
5. Well written with clear presentation [Reviewer 99Gn] [Reviewer jTir]
6. Be applicable to more wide models and datasets [Reviewer LMJF]
7. Good clarity [Reviewer jTir]

And following a thorough consideration of the valuable suggestions put forth by the reviewers, we have modified some priority points of our paper for improvement and more clear clarification.

+ Modify presentation of abstract and introduction [Reviewer LMJF].
+ Modification of Equation 10 and Equation 12 [Reviewer LMJF]
+ Modification of the proof of Theorem 2.1 [Reviewer 99Gn]
+ Add Figure 4 in Appendix [Reviewer 99Gn]

Due to the time limitation, we will improve other points in subsequent versions.  We thank all reviewers again for their thoughtful reviews and for spending time writing the reviews.

---

### Author Response · Authors · 2024-11-28

Dear all reviewers,

We sincerely appreciate your time and efforts in reviewing our paper. As the deadline for the author-reviewer discussion period is approaching, we kindly ask you to check our latest response if you have not checked.  We hope that our response has adequately addressed your concerns. And if you have additional questions, please don't hesitate to let us know, we are willing to continue our communication in the last few days.

Best Regards,

Authors of paper3161

---

### Meta-Review · Area_Chair_LEJk · 2024-12-20

**Metareview:**

This paper focuses on graph reasoning tasks and combines graph neural networks and language models to model both the structure and text information. It adopts an alternating training scheme to optimize the GNN and the PLM. The proposed method demonstrates superior results to baselines.

This paper receives a rather borderline score of 5,5,6,6. I read all the reviews and the author responses, and I think the main merits of the paper is on the empirical side. Empirical contribution alone has no problem at ICLR, but the paper seems to emphasize the proposed alternative training scheme significantly as methodological novelty – which is perceived as one of the main strengths by multiple reviewers. I however echo Reviewer LMJF’s comment that such an alternative training scheme has been existing in several previous works on combining GNNs and PLMs, and this paper is more like applying a similar alternative training scheme to new graph reasoning tasks. Therefore, I think the paper is not ready to be published at ICLR.

**Additional Comments On Reviewer Discussion:**

The reviewers mainly raise points on more experiments for various backend LLMs, complexity of the training paradigm, theoretical derivations, method novelty, and some clarification issues. The authors have done a good job on addressing most of the concerns and some reviewers updated their scores correspondingly. The method novelty raised by Reviewer LMJF is the most concerning to me.

---

### Decision · Program_Chairs · 2025-01-22

Reject